# Diagnostic Value of Urine Cytology in Pharmacologically Forced Diuresis for Upper Tract Urothelial Carcinoma Diagnosis and Follow-Up

**DOI:** 10.3390/cancers16040758

**Published:** 2024-02-12

**Authors:** Nicola Giudici, Jennifer Blarer, Niranjan Sathianathen, Fiona C. Burkhard, Patrick Y. Wuethrich, George N. Thalmann, Roland Seiler, Marc A. Furrer

**Affiliations:** 1Department of Urology, Hospital Center Biel, Vogelsang 84, 2501 Biel, Switzerland; nicolagiudici@gmail.com (N.G.); jennifer.seiler-blarer@szb-chb.ch (J.B.); r_seiler@gmx.ch (R.S.); 2Department of Urology, The University of Melbourne, Royal Melbourne Hospital, Parkville, VIC 3052, Australia; niranjan19@gmail.com; 3Department of Urology, Inselspital, Bern University Hospital, University of Bern, 3010 Bern, Switzerland; fiona.burkhard@insel.ch (F.C.B.); george.thalmann@insel.ch (G.N.T.); 4Department of Anaesthesiology and Pain Medicine, Inselspital, Bern University Hospital, University of Bern, 3010 Bern, Switzerland; patrick.wuethrich@insel.ch; 5Department for BioMedical Research, Translational Organoid Resource Core, University of Bern, 3010 Bern, Switzerland; 6Department of Urology, Solothurner Spitäler AG, Kantonsspital Olten, 4600 Olten, Switzerland; 7Department of Urology, Solothurner Spitäler AG, Bürgerspital Solothurn, 4500 Solothurn, Switzerland

**Keywords:** diagnostic value for diagnosis and follow-up, urine cytology, pharmacologically forced diuresis, upper tract urothelial carcinoma

## Abstract

**Simple Summary:**

The diagnosis of primary upper urinary tract urothelial carcinoma is challenging and may necessitate invasive procedures. In this study, through analysing two cohorts, we evaluated the diagnostic accuracy of drug-induced turbulence in the upper urinary tract and non-invasive urine sampling in the diagnosis of primary upper tract urothelial carcinoma and upper tract urothelial carcinoma recurrence during follow-up after radical cystectomy for primary bladder cancer. As a diagnostic tool, cytology of pharmacologically forced diuresis performs better in patients with invasive upper tract urothelial carcinoma and concomitant carcinoma in situ. As a method of surveillance, positive cytology of pharmacologically forced diuresis was related to cancer recurrence and could even detect recurrence in the urethra in cases involving an orthotopic bladder substitute. In conclusion, urine cytology in pharmacologically forced diuresis may be useful in patients with suspected upper tract urothelial carcinoma, especially in cases with contraindications for imaging or when achieving endoscopic access to the upper urinary tract is difficult.

**Abstract:**

We performed a urine cytology analysis of a pharmacologically induced diuresis for the diagnosis of upper tract urothelial carcinoma. To evaluate the diagnostic value of cytology of pharmacologically forced diuresis, an initial cohort of 77 consecutive patients with primary upper tract urothelial carcinoma treated via radical surgery was enrolled. To evaluate pharmacologically forced diuresis cytology as a follow-up procedure, a second cohort of 1250 patients who underwent a radical cystectomy for bladder cancer was selected. In the first cohort, the sensitivity of cytology of pharmacologically forced diuresis in patients with invasive, high-grade, low-grade, and concomitant carcinoma in situ was 8%, 9%, 0%, and 14%, respectively. In the second cohort, cytology of pharmacologically forced diuresis was positive in 30/689 (4.3%) patients, in whom upper urinary tract recurrence was present in 21/30 (70%) of cases, and urethral recurrence was present in 8/30 (26%) of cases. As a follow-up tool, cytology of pharmacologically forced diuresis showed a sensitivity, specificity, and positive and negative predictive values of 60%, 99%, 70%, and 98%, respectively. Overall, as a diagnostic tool, the sensitivity of cytology of pharmacologically forced diuresis is slightly better in patients with invasive upper tract urothelial carcinoma and concomitant carcinoma in situ. As a follow-up method, positive cytology of pharmacologically forced diuresis is strongly related to cancer recurrence and can reveal urethral recurrence. Cytology of pharmacologically forced diuresis might be useful in cases with contraindications for imaging or when achieving endoscopic access to the upper urinary tract is difficult.

## 1. Introduction

Urothelial carcinomas are the 11th most common cancers in developed countries [1,2]. Upper tract urothelial carcinoma constitutes a relatively uncommon subset of urothelial carcinoma, making up 5–10% of all cases of urothelial carcinoma and being three times more common in men compared to women. Among upper tract urothelial carcinomas, pyelocaliceal tumours are twice as prevalent as ureteral tumours. Concurrent bladder tumours are identified in 17% of individuals with upper tract urothelial carcinoma. Notably, bladder recurrence after upper tract urothelial carcinoma is prevalent, manifesting in 22–47% of patients, while contralateral upper tract recurrence is less frequent, occurring in only 2–6% of cases [3,4,5,6]. The most common symptoms in patients with upper tract urothelial carcinoma are haematuria (70–80%), followed by flank pain (20–30%) [7,8,9], followed by the presentation of a lumbar mass, occurring in 10–20% of patients. As such, approximately half of patients are asymptomatic at presentation [10], and their case(s) may only become apparent in the event of symptoms due to locally advanced disease symptoms such as malaise and fatigue [10,11,12,13].

On the other hand, upper tract recurrence following radical cystectomy for a primary bladder urothelial carcinoma is deemed rare, typically occurring 24–36 months post operation, with an incidence ranging from 0.8 to 6.4% [14,15,16]. Established risk factors are carcinoma in situ at radical cystectomy, tumour multifocality and a history of multifocal bladder urothelial carcinoma, a prior history of upper tract urothelial carcinoma prior to radical cystectomy, and a positive ureteral or urethral margin [14,15,17]. 

The diagnosis of primary upper tract urothelial carcinoma is challenging and mainly based on invasive procedures. Bladder urinary cytology has a relatively low sensitivity for the diagnosis of upper tract urothelial carcinoma and has no role in the diagnosis of upper tract urothelial carcinoma. Invasive procedures such as selective cytology from the upper urinary tract show a sensitivity of 70–75% for high-grade upper tract urothelial carcinoma [18]. Not surprisingly and in line with what has been observed for bladder tumours, performance seems to be improved by the turbulence of liquids in the upper urinary tract. E-cadherin is a cell–cell adhesion transmembrane glycoprotein and a core component of epithelial junctions. A decrease in E-cadherin expression has been described as a prognostic factor and is associated with tumour aggressiveness in upper tract urothelial carcinoma. In this context, using a mechanical force to obtain a cytology in the upper urinary tract is critical to increase diagnostic performance [19]. In a prospective study, cytology of liquids obtained through barbotage outperformed standard cytology of the upper urinary tract (sensitivity: 92% for high-grade; 87% for low-grade) [20]. The drawback of urine cytology obtained by barbotage is the need for invasive intervention. Therefore, we hypothesized that through a pharmacologically induced diuresis and given the tendency of upper tract urothelial carcinomas to exhibit fragile adhesion molecules, the generation of turbulences in the upper urinary tract and sampling of voided urine thereafter may be a promising approach for enhancing biomarker performance. 

The aim of this study was to evaluate urine cytology analyses of a pharmacologically forced diuresis in the diagnostic setting of two different cohorts, namely patients with primary upper tract urothelial carcinoma and urothelial cancer of the bladder, respectively. 

## 2. Materials and Methods

### 2.1. Study Population

As an initial cohort, we retrospectively enrolled 77 consecutive patients from 2009 to 2022 with confirmed upper tract urothelial carcinoma treated with radical nephroureterectomy or segmental ureterectomy with a curative intent. Patients with upper tract urothelial carcinoma who underwent endoscopic treatments as well as patients with previous radical cystectomy were excluded. Cytology of pharmacologically forced diuresis was performed at discretion of the treating urologist as routine practice in tumour-naive patients in case of radiological suspicion for upper tract urothelial carcinoma to support the diagnostic workup in our institution. In 25/77 (35%) patients, cytology of pharmacologically forced diuresis was performed preoperatively. Postoperative follow-up was performed in accordance with the European guidelines [21].

A declaration of consent of the University Hospital of Bern (Inselspital) for the use of biological material and health-related data for medical research was signed by each patient.

A second retrospective cohort (*n* = 1250) of patients who underwent a radical cystectomy (with or without neoadjuvant chemotherapy) according to a standard protocol and with a curative intent due to urothelial cancer of the bladder from 2000 to 2020 was also included in our study. In 689/1250 (55%) patients, cytology of pharmacologically forced diuresis was performed during follow-up as routine practice in our institution every six months in the first two years post operation and yearly thereafter. Given that the majority of patients had underwent several cytological examinations at different time points during follow-up, a total of 1431 cytology specimens had been analysed. Preoperative investigation, surgical techniques such as radical cystectomy, pelvic lymph node dissection, and urinary diversion [22,23,24,25,26]. Bedside ultrasound imaging of the upper urinary tract with CT or MRI urography was performed at 6, 12, 18 and 24 months after radical cystectomy. CT urography was the modality of choice. MR-urography was performed only in the case of contraindications for CT, given its slightly lower accuracy [27]. In addition, cytology of pharmacologically forced diuresis is performed as a follow-up control to detect recurrence in the upper urinary tract at 6 months and yearly thereafter in cases where the tumour is located close to or in the ureter, cases of multifocal carcinoma in situ in the cystectomy specimen, and cases of histopathologically confirmed lymph node metastasis at the time of radical cystectomy, or on demand in cases of suspicious upper urinary tract imaging. 

The collection of the data of this second cohort was carried out in accordance with the Strengthening the Reporting of Observational Studies in Epidemiology (STROBE) guidelines. Ethical approval for this study (Ethical Committee No KEKBE 2016-00660) was provided by the Ethical Committee of Canton Bern, Switzerland (Chairperson Professor C. Seiler), on 2 June 2016, and the need for informed consent was waived.

### 2.2. Cytology of Pharmacologically Forced Diuresis: Technique

Cytology of pharmacologically forced diuresis was performed by collecting urine after a forced diuresis induced by oral furosemide administration (40 mg) and an oral fluid overload. In an outpatient setting, patients were asked to collect approximately 500 mL of spontaneously voided urine after consuming the diuretic and an oral water intake of 1 L (i.e., forced diuresis). The urine samples were then stored in a fridge (at 4 °C) and processed at the Department of Pathology of the University Hospital Bern within 24 h (see Figure 1).

### 2.3. Pathologic and Cytological Analysis

All specimens after radical cystectomy and radical nephroureterectomy or segmental ureterectomy, respectively, as well as collected urine specimens, were evaluated by specialized genitourinary pathologists. Cytology was classified as either high-grade, atypical/suspicious/low-grade, or negative according to the Paris System for reporting urinary cytology [28]. The pathologies of the surgical specimens were analysed according to the histological classification published by WHO and the International Society of Urological Pathology in 2016 [29].

### 2.4. Statistical Analysis 

For both cohorts, we used descriptive analysis. In the first cohort, the sensitivity of pharmacologically forced diuresis cytology was measured for the entire population as well as for subgroups (invasive and high-grade cancer and concomitant carcinoma in situ). In the second cohort, the sensitivity, specificity, and positive and negative predictive value of pharmacologically forced diuresis cytology were evaluated during postoperative follow-up using descriptive analysis for the entire population as well as for subgroups (muscle-invasive disease, non-muscle-invasive disease, lymph node-positive disease, and concomitant carcinoma in situ). The performance of pharmacologically forced diuresis cytology between different subgroups was compared using Youden’s index, a chi-squared test, and Fisher’s Exact Test for categorical variables, and a *p*-value of less than 0.05 was considered significant. Statistical analysis was performed using IBM SPSS^®^ v. 25 statistical software (SPSS Inc., Chicago, IL, USA).

### 2.5. Figures

All figures presented herein were created using BioRender.com.

### 2.6. Definition of Positivity

We assessed the diagnostic value of pharmacologically forced diuresis cytology and used two different criteria (“stringent” and “soft”) to define positive cytology (Table 1). In the “stringent” criterion group, positivity was only defined in the presence of high-grade UC (HGUC) according to the Paris System [28]. The other categories—“atypical urothelial cells (AUC)”, “suspicious for high-grade UC (suspicious)”, and “low-grade urothelial neoplasia (LGUN)”—were considered “negative” in the “stringent” criterion group but positive in the “soft” criterion group. Finally, “negative for high-grade UC (negative)” was considered negative for both criteria. 

## 3. Results

### 3.1. First Cohort—Clinicopathological Characteristics 

For the clinicopathological characteristics of the first cohort (*n* = 25), see Table 2. The median age of patients (8 females, 17 males) at surgery was 66 years. A history of tobacco consumption was noted in 13/25 (52%) patients. At first diagnosis, hydronephrosis was found in 13/25 (52%) patients. Tumours were located in the pelvic–calyceal system in 15/25 (60%) patients and in the ureter in 7/25 (28%) patients. Multifocal upper tract urothelial carcinoma was diagnosed in 3/25 (12%) patients, and concomitant carcinoma in situ was diagnosed in 8/25 (32%) patients. Ureteroscopic biopsy as an integral part of initial diagnosis was performed in 12/25 (48%) cases, and local cytology (barbotage and non-barbotage) was performed in 20/25 (80%) of cases, of which high-grade, atypical/suspicious/low-grade, and negative urothelial cells were present in 7/25 (28%), 9/25 (36%), and 4/25 (16%) of cases, respectively. Muscle-invasive upper tract urothelial carcinoma in the radical nephroureterectomy or segmental ureterectomy specimens was found in 17/25 (57%) patients [21]. The median follow-up was 24.5 months, with extra-vesical recurrence occurring in 28% (42% local, 29% nodal, and 29% distant) of patients. Bladder recurrence during follow-up was diagnosed in 15/25 (60%) patients after a median of 6 months (range 3–53). Finally, 3-year overall and cancer-specific survival were 71% and 82%, respectively.

### 3.2. Diagnostic Value of Cytology of Pharmacologically Forced Diuresis 

Cytology of pharmacologically forced diuresis was positive, with high-grade urothelial carcinoma being found in 2/25 (8%) patients. Atypia/suspicious/low-grade and negative cytology of pharmacologically forced diuresis was diagnosed in 6/25 (24%) and 17/25 (68%) patients (see Figure 2).

The sensitivity of pharmacologically forced diuresis cytology, according to the stringent criterion, in patients with invasive upper tract urothelial carcinoma, high-grade UC, low-grade UC, and concomitant carcinoma in situ was 8%, 9%, 0%, and 14%, respectively. 

According to the soft criterion, sensitivity in patients with invasive upper tract urothelial carcinoma, high-grade and low-grade upper tract urothelial carcinoma, and concomitant carcinoma in situ was 30%, 32%, 50%, and 25%, respectively. 

Interestingly, among the 23 patients with negative cytology of pharmacologically forced diuresis, 12/23 (52%) had muscle-invasive upper tract urothelial carcinoma (pT-stage ≥ 2), 7/23 (30%) had a concomitant carcinoma in situ, and 5/23 (22%) had a subsequent high-grade selective urine cytology. Of the two patients with low-grade urothelial carcinoma, one presented atypia/suspicious/low-grade cytology of pharmacologically forced diuresis.

### 3.3. Second Cohort—Clinicopathological Characteristics

For the clinicopathological characteristics of the second cohort (*n* = 689), see Table 3. The median age of patients (201 [29%] females; 488 [71%] males) at radical cystectomy was 68 years. A history of tobacco consumption was noted in 405/689 (59%) patients. At diagnosis, hydronephrosis was found in 131/689 (19%) patients. Overall, 41/689 (6%) patients had upper tract urothelial carcinoma prior to or at the time (synchronous) of radical cystectomy. Muscle-invasive disease was diagnosed in 510/689 (74%) of radical cystectomy specimens. High-grade bladder urothelial carcinoma was found in 625/689 (91%) patients. Regarding extended pelvic lymph node dissection, the median number of removed lymph nodes was 33 (IQR 24–43), with lymph node metastasis being found in 161/689 (23%) cases. The positive surgical margin rate was 2.8% (19/689). During follow-up, local recurrence and distant metastasis occurred in 92/689 (13%) and 177/689 (26%) patients.

### 3.4. Diagnostic Value of Pharmacologically Forced Diuresis Cytology

To analyse the second cohort, we used two different approaches. The first involved retrospectively evaluating data obtained from patients who underwent a radical cystectomy due to urothelial cancer of the bladder in whom pharmacologically forced diuresis cytology was performed during follow-up; the second approach involved analysing all patients in the same cohort starting from those in whom upper urinary tract recurrence was confirmed during follow-up.

During the follow-up periods of the 689 patients of the second cohort, a total of 1431 cytologies of pharmacologically forced diuresis were analysed. Of them, 3.4% (49/1431) contained urothelial cancer cells that were considered as ‘positive’ according to the stringent criterion (Table 1). Cytology of pharmacologically forced diuresis was positive in 30/689 (4.3%) patients. In those patients, upper urinary tract and urethral recurrence occurred in 21/30 (70%) and 8/30 (26.7%), respectively. One patient (3.3%) had a false-positive cytology of pharmacologically forced diuresis.

We then analysed the same cohort using another approach, starting from patients with proven recurrence in the upper urinary tract. After radical cystectomy, recurrence in the upper urinary tract was diagnosed in 3.7% (46/1250) of patients during follow-up. In 31/46 (67%) patients, cytology of pharmacologically forced diuresis was analysed prior to treatment (of the upper urinary tract recurrence), while urothelial cancer cells (true-positive) were found in 17/31 (55%) tests. In 2/46 (4%) patients, cancer recurrence in the upper urinary tract was detected by cytology of pharmacologically forced diuresis only, while upper urinary tract imaging was negative (Table 4). 

As such, when assessing the diagnostic accuracy pharmacologically forced diuresis cytology according to the stringent criterion, the sensitivity, specificity, and positive and negative predictive values were 60% (42–76%), 99% (97–99%), 70% (50–85%), and 98% (96–99%), respectively, with a Youden’s index of 0.59 (0.42–0.75); see Table 5.

Due to its nature, a high number of atypical findings in forced diuresis samples are found in patients with an ileal conduit. Therefore, the diagnostic accuracy of pharmacologically forced diuresis cytology according to the soft criterion was not evaluated.

### 3.5. Clinical Parameters Associated with a Higher Accuracy of Cytology of Pharmacologically Forced Diuresis in Both Cohorts 

No perioperative factor was statistically significant in predicting a positive cytology of pharmacologically forced diuresis in the primary diagnosis of upper tract urothelial carcinoma, but a trend was observed. The preoperative factors associated with a positive cytology of pharmacologically forced diuresis were smoking history (*p* = 0.07), positive selective ureter cytology (*p* = 0.07) at time of initial diagnostic workup, normal preoperative kidney function > 75 mL/min/1.73 m^2^ (*p* = 0.09), absence of hydronephrosis at diagnosis (*p* = 0.22), and variant histology (*p* = 0.08). 

Similarly, in the second cohort, no preoperative factor was significantly associated with a higher sensitivity and specificity for cytology of pharmacologically forced diuresis (see Table 6 and Table 7).

## 4. Discussion

In this study, we evaluated the diagnostic accuracy and clinical significance of drug-induced turbulence in the upper urinary tract and subsequent urine sampling in the diagnosis of primary upper tract urothelial carcinoma and upper tract urothelial carcinoma recurrence during follow-up after radical cystectomy for primary bladder urothelial carcinoma. 

Not surprisingly for primary upper tract urothelial carcinoma, the performance of cytologies of pharmacologically forced diuresis was only adequate in the case of invasive and high-grade pathology or concomitant carcinoma in situ. 

Similarly, in the second cohort, precursor lesions of advanced tumours such as non-muscle-invasive disease and carcinoma in situ were associated with better-sensitivity cytology of pharmacologically forced diuresis. This notable difference between the two cohorts is partly attributable to upper urinary tract recurrence being detected at the early stages in the second cohort due to intensive follow-up after radical cystectomy. 

In fact, about 60% of patients with confirmed recurrence in the upper urinary tract had non-muscle-invasive disease in their radical cystectomy specimens, while this was the case in less than 50% of the patients in the first cohort. In the second cohort, the negative predictive and specificity values were remarkably high, likely due to the relatively low rate of recurrence in the upper urinary tract after radical cystectomy. Some preoperative factors may interfere with the performance of pharmacologically forced diuresis cytology (e.g., normal kidney function and smoking history). However, none of these parameters reached statistical significance. Interestingly, all patients with a positive cytology of pharmacologically forced diuresis in primary upper tract urothelial carcinoma developed a bladder recurrence. Apparently, patients with cancer cell desquamation are at risk of bladder recurrence, which may support the hypothesis that tumour cells can spread from the upper urinary tract to the bladder [30]. 

Overall, the performance of pharmacologically forced diuresis cytology for the follow-up of issues in the upper urinary tract after radical cystectomy (second cohort) was better than in the first cohort. Also, due to the relatively high rate of false-negative results, cytology of pharmacologically forced diuresis cannot replace standard diagnostic techniques (i.e., CT and MR imaging or ureteroscopic biopsy). However, in a few cases, it allowed for the diagnosis of recurrence in the upper urinary tract despite negative imaging, and in cases involving an orthotopic bladder substitute, it could diagnose recurrence in the urethra during follow-up.Cytology of pharmacologically forced diuresis is a simple and non-invasive diagnostic tool. However, the current diagnostic gold standard for upper tract urothelial carcinoma is predominantly based on CT and MR imaging and invasive tumour biopsy or cytology. A major limitation of cytology of pharmacologically forced diuresis is that cytological results in patients with ureteral tumours obstructing urine flow are associated with false-negative test results. This is not surprising given that the tumour cells are not washed out. As such, negative predictive factors (i.e., hydronephrosis) allow for the negative selection of patients for testing.

To our knowledge, this is the first study to assess the diagnostic utility of drug-induced turbulence in the upper urinary tract and urine cytology in the context of upper tract urothelial carcinoma. However, the use of biomarkers in the setting of urothelial carcinoma is a constantly evolving field. Currently, numerous other molecular biomarkers in the setting of upper tract urothelial carcinoma are under investigation. Biomarkers currently under investigation include circulating tumour cells, circulating tumour DNA and exfoliated cells in urine, proteins, mRNAs, miR-NAs, long noncoding RNAs, and vesicles [31]. Studies have evaluated tumour-associated cellular antigens (ImmunoCyt/uCyt+) [32], chromosomal anomalies (UroVysion) [33], nuclear matrix protein-22 [34], hCFHrp (BTA stat) [35], DNA analysis, and genetic mutations (EpiCheckTM) [36,37,38,39]. 

Considering the morbidity statistics and limitations of the currently recommended diagnostic workup, non-invasive tumour biomarkers are promising. To date, despite promising results, performance and accuracy have been observed to vary significantly (sensitivity: 44–91%; specificity 67–100%) and have been analysed mostly in single studies with small populations [31,32,33,34,35,36,37,40]. 

Nevertheless, cytology of pharmacologically forced diuresis may have implications for clinical practice. As such, cytology of pharmacologically forced diuresis might be considered to support the diagnostic workup for upper tract urothelial carcinoma in well-selected patients (e.g., in the follow-up of patients with associated risk factor for recurrence in the upper urinary tract who underwent radical cystectomy for bladder urothelial carcinoma). Clearly, increased prevalence in a specified population increases the positive predictive value. However, the use of cytology of pharmacologically forced diuresis in the primary diagnosis of upper tract urothelial carcinoma seems limited by its low diagnostic performance, and even in case of positive results it does not allow for the avoidance of the standard diagnostic assessment. On the other hand, cytology of pharmacologically forced diuresis is useful in cases with contraindications for CT or MR urography (e.g., allergy to contrast agents, chronic renal insufficiency, claustrophobia) or when endoscopic access to the upper urinary tract is difficult. As such, a long ileal conduit or mucosal folds in the proximity of the neo-orifices may impede endoscopic investigation. In these situations, cytology of pharmacologically forced diuresis may add diagnostic information to other baseline investigations (i.e., clinical examination, non-contrast CT, ultrasound).

Also, it remains an open question whether the use of cytology of pharmacologically forced diuresis may reduce the performance of invasive procedures or imaging. For these reasons, the need for better-performing biomarkers remains. 

Before the implementation of biomarker testing, the economic burdens and benefits for patients need to be weighed up against treatment costs. We therefore evaluated the costs of cytology of pharmacologically forced diuresis in our department and weighed them against others. In comparison to other biomarkers, the costs of cytology of pharmacologically forced diuresis analysis in our institution are USD 87 in an outpatient setting. Costs for the currently available biomarkers are variable, ranging between USD 25 and USD 800 [31]. However, the values of new biomarkers for daily, routine use also needs to be balanced against their costs and validated based on several datasets.

The results of this study must be interpreted in the context of its limitations. The retrospective nature of this study may have introduced selection bias. The first cohort had a limited sample size, lowering the statistical power of the study. Also, predictive values for the first cohort were not calculated due to the initial patient selection criteria. In addition, despite processing the samples as early as possible, the collection and processing of the cytology of pharmacologically forced diuresis samples was not controlled. As such, errors and delays in both steps may have acted as sources of bias. Furthermore, given that all patients underwent pharmacologically forced diuresis, a comparative analysis with patients followed-up with a conventional cytology was not possible. We can only speculate that pharmacologically forced diuresis obtains a larger number of cells, which may increase its diagnostic accuracy. In this context, Brimo et al. reported that a preliminary cut-off of 10 cells appears to be easily applicable and valid from a clinical standpoint when assigning cases with atypical urothelial cells to the “positive” or “suspicious” categories [41].

## 5. Conclusions

Urine cytology derived from forced diuresis is a simple and non-invasive diagnostic method.

In a diagnostic context, urine cytology derived from forced diuresis has a low accuracy, and our results do not support its use in unselected cases. As a follow-up method, positive cytology of pharmacologically forced diuresis was strongly related to cancer recurrence and could even reveal cancer recurrence in the urethra in cases with an orthotopic bladder substitute. While cytology of pharmacologically forced diuresis may add diagnostic value in a few cases with recurrence of carcinoma in situ, its diagnostic accuracy is not adequate as a standalone test for recurrence. Therefore, there is a need for more reliable biomarkers for the follow-up and diagnosis of upper tract urothelial carcinoma. Furthermore, cytology of pharmacologically forced diuresis is useful in cases involving contraindications for CT or MR urography or when achieving endoscopic access to the upper urinary tract is difficult.

## Figures and Tables

**Figure 1 cancers-16-00758-f001:**
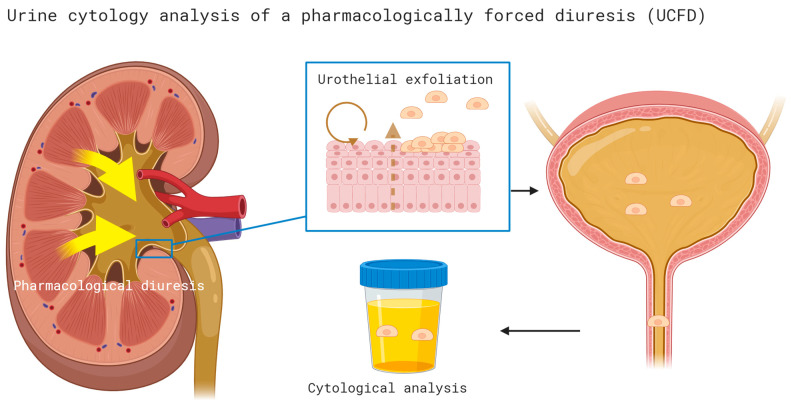
Cytology of pharmacologically forced diuresis was performed by collecting urine after a forced diuresis induced by oral furosemide administration (40 mg) and an oral fluid overload. In an outpatient setting, patients were asked to collect approximately 500 mL of spontaneously voided urine after consuming the diuretic and an oral water intake of 1 L (i.e., forced diuresis). The urine samples were then stored in a fridge (at 4 °C) and processed at the Department of Pathology of the University Hospital Bern within 24 h.

**Figure 2 cancers-16-00758-f002:**
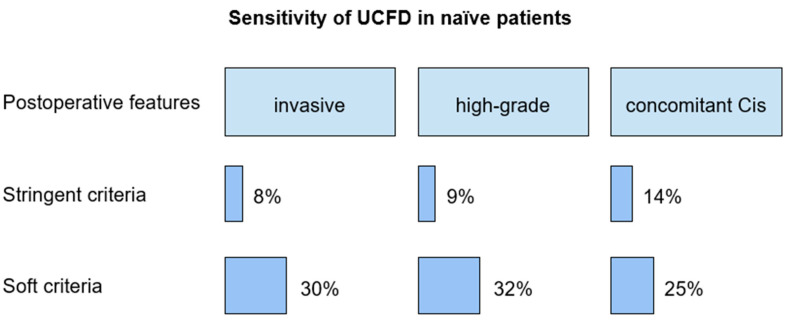
Diagnostic value of UCFD in the first cohort. UCFD, urine cytology analysis of a pharmacologically forced diuresis; Cis, carcinoma in situ.

**Table 1 cancers-16-00758-t001:** Definition of positivity: “stringent” and “soft” criteria.

	High-Grade UC	Atypical Urothelial Cells, Suspicious for High-Grade UC, Low-Grade Urothelial Neoplasia	Negative for High-Grade UC
Stringent criteria	+	-	-
Soft criteria	+	+	-

UC, urothelial cancer.

**Table 2 cancers-16-00758-t002:** Clinicopathological characteristics of the first cohort.

Patient data (*n* = 25)			
	Age (median, range) at surgery [years]		66 (81–44)
	Female/male, *n* (%)		8 (32%)/17 (68%)
	Follow-up (median, range) [months]		24.5 (2–95)
Diagnostic data			
	Pelvic–calyceal/ureter location, *n* (%)		15 (60%)/7 (28%)
	Multifocality, *n* (%)		3 (12%)
	Hydronephrosis, *n* (%)		14 (56%)
	Selective cytology, *n* (%)		20 (80%)
		High-grade/atypia, suspicious, low-grade/negative	7 (35%)/9 (45%)/4 (20%)
	URS biopsy, *n* (%)		12 (48%)
Surgery data			
	≤pT2/>pT2, *n* (%)		12 (48%)/13 (52%)
	Concomitant carcinoma in situ, *n* (%)		8 (32%)
	High-grade/Low-grade, *n* (%)		23 (92%)/2 (8%)
	pN0/pN+, *n* (%)		20 (80%)/5 (20%)
	Positive surgical margins, *n* (%)		1 (4%)
Chemotherapy			
	Neoadjuvant/adjuvant, *n* (%)		0 (0%)/1 (4%)
Recurrence			
	Extra-vesical/vesical, *n* (%)		7 (28%)/15 (60%)
Outcomes			
	3 yr overall survival (%)		71
	3 yr cancer-specific survival (%)		82

**Table 3 cancers-16-00758-t003:** Clinicopathological characteristics of the second cohort.

Patient data (*n* = 689)		
	Age (median, range) at surgery [years]	68 (30–89)
	Female/male, *n* (%)	201 (29%)/488 (71%)
	Follow-up (median, range) [months]	37.5 (0.3–212)
Diagnostic data		
	Synchronous UTUC, *n* (%)	5 (1%)
	Metachronous UTUC prior to RC, *n* (%)	36 (5%)
	Hydronephrosis, *n* (%)	131 (19%)
Surgery data		
	≤pT2/>pT2, *n* (%)	510 (74%)/179 (26%)
	High-grade/Low-grade	625 (91%)/64 (9%)
	Concomitant carcinoma in situ, *n* (%)	280 (40%)
	pN0/pN+, *n* (%)	528 (77%)/161 (23%)
	Median lymph nodes removed (median, IQR)	33 (24–43)
	Positive surgical margins, *n* (%)	19 (3%)
Chemotherapy		
	Neoadjuvant/adjuvant, *n* (%)	118 (17%)/70 (10%)
Recurrence		
	UTUC/local/distant, *n* (%)	46 (7%)/92 (13%)/177 (26%)
Outcomes		
	3-year overall survival (%)	68
	3-year cancer-specific survival (%)	90
	5 year overall survival (%)	56
	5 year cancer-specific survival (%)	76

UTUC, upper tract urothelial cancer.

**Table 4 cancers-16-00758-t004:** Diagnostic workup, treatment, and histopathological features of 46 patients with proven upper tract recurrence after radical cystectomy for urothelial bladder cancer.

Diagnostic Workup		*n* = 46
Forced diuresis performed, *n* (%)		31 (67)
True-positive, *n* (%)		17 (55)
False-negative, *n* (%)		14 (45)
Diagnostic tool revealing UUT recurrence		
	Forced diuresis only, *n* (%)	2 (4)
	Forced diuresis and CT, *n* (%)	13 (28)
	Forced diuresis and invasive workup	2 (4)
	(selective ureteral cytologies and ureteroscopy with biopsy)	
	No forced diuresis performed	15 (33)
	CT, *n* (%)	27 (59)
	Invasive workup	1 (2)
	(selective ureteral cytologies and ureteroscopy with biopsy)	
	Incidental finding (resection of uretero-ileal anastomotic stricture)	1 (2)
Treatment		*n* = 46
	BCG only	4 (9)
	BCG and subsequent NUT (for second recurrence)	1 (2)
	Laser treatment and adjuvant farmorubicin instillation	6 (13)
	Laser treatment and subsequent nephroureterectomy (for second recurrence)	2 (4)
	NUT and lymph node dissection	15 (33)
	Distal ureteric/ureteroileal anastomotic resection	4 (9)
	Palliative chemotherapy	9 (20)
	Palliative radiotherapy	1 (2)
	Best supportive care	4 (9)
Histopathological features		*n =* 44
T stage		
	Ta	10 (23)
	T1	8 (18)
	T2	3 (7)
	T3	14 (32)
	T4	1 (2)
	CIS only	8 (18)
Grade		
	G1	2 (4)
	G2	8 (18)
	G3	34 (77)

CIS, carcinoma in situ.

**Table 5 cancers-16-00758-t005:** Diagnostic value of UCFD in the second cohort according to the stringent criterion.

	Entire Cohort (*n* = 689)	Muscle-Invasive Disease (*n* = 510)	Non-Muscle-Invasive Disease (*n* = 179)	Concomitant Carcinoma In Situ (*n* = 280)
Sensitivity	60% (42–76%)	50% (31–69%)	82% (48–97%)	75% (51–90%)
Specificity	99% (97–99%)	99% (98–100%)	98% (94–99%)	98% (95–99%)
PPV	70% (50–85%)	71% (44–89%)	69% (39–90%)	75% (51–90%)
NPV	98% (96–99%)	98% (96–99%)	99% (95–100%)	98% (95–99%)
Youden’s index	0.59 (0.42–0.75)	0.49 (0.29–0.69)	0.79 (0.57–1)	0.73 (0.54–0.92)

UCFD, urine cytology analysis of a pharmacologically forced diuresis; PPV, positive predictive value; NPV, negative predictive value.

**Table 6 cancers-16-00758-t006:** Univariate analysis of the effect of clinical parameters on the predictive value of UCFD according to stringent criteria in the diagnosis of primary UTUC.

		UCFD-Positive (*n* = 2)	UCFD-Negative (*n* = 23)	*p* Value
Preoperative				
	Hydronephrosis (*n*)	0 (0%)	13 (56%)	0.22
	Smoking history positive (*n*)	2 (100%)	5 (22%)	0.07
	Pelvic–calyceal/ureter location (*n*)	2 (100%)/0 (0%)	13 (56%)/7 (30%)	0.50/1.00
	Selective ureter cytology positive (*n*)	2 (100%)	5 (22%)	0.07
	Preoperative eGFR > 75 mL/min/1.73 m^2^	2 (100%)	6 (26%)	0.09
	Variant histology	1 (sarcomatoid) (50%)	0 (0%)	0.08
Postoperative				
	pN+	1 (50%)	4 (17%)	0.36
	Bladder recurrence	2 (100%)	13 (56%)	0.50

UCFD, urine cytology analysis of a pharmacologically forced diuresis.

**Table 7 cancers-16-00758-t007:** Univariate analysis for the sensitivity and specificity of UCFD in the diagnosis of upper urinary tract recurrence after radical cystectomy and urinary diversion.

	Univariate Analysis for Sensitivity (*p*-Value)	Univariate Analysis for Specificity (*p*-Value)
Lymph node metastasis	0.48	0.36
Muscle invasion	0.16	0.36
Positive surgical margin	0.39	>0.99
Nephrostomy insertion preoperatively	0.26	0.93
Stent insertion preoperatively	0.39	0.85
Upper urinary tract cancer prior to radical cystectomy	0.06	0.65
Recurrent bladder cancer prior to radical cystectomy	0.26	>0.99

UCFD, urine cytology analysis of a pharmacologically forced diuresis.

## Data Availability

The data presented in this study are available from the corresponding author upon request.

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
