# Peer review of "Diagnostic Value of Urine Cytology in Pharmacologically Forced Diuresis for Upper Tract Urothelial Carcinoma Diagnosis and Follow-Up"

_cancers, 2024, doi:10.3390/cancers16040758_

Round 1

Reviewer 1 Report

Comments and Suggestions for Authors

The presentation of the data is confusing. The description of the second cohort is inconsistent, with n=1250 mentioned at one point and n=689 at another. Additionally, on page nine, it is stated that a total of 1431 cytology tests from pharmacologically forced diuresis were analyzed. Furthermore, in cohort one, 25 individuals underwent preoperative cytology examination, and the positive rate obtained, regardless of Stringent or soft criteria, was very low. The reviewer acknowledges the significance of urine cytology analysis of a pharmacologically induced diuresis and is open to accepting both positive and negative results. However, the conclusion cannot be drawn from such a confusing presentation of the data.

Author Response

 Reviewer #1

  1. The presentation of the data is confusing. The description of the second cohort is inconsistent, with n=1250 mentioned at one point and n=689 at another. Additionally, on page nine, it is stated that a total of 1431 cytology tests from pharmacologically forced diuresis were analyzed.

Reply: We greatly appreciate this important objection. The discrepancy between the two cohorts is due to the fact that of the n=1250 patients initially analyzed, only 689 underwent a cytological examination of a forced diuresis during the follow-up. We agree with the reviewer that as currently presented may be confusing, and therefore we have adapted the formulation in the methods section to allow for a better understanding, see page 4, paragraph 3. However, we think it is important to point out that the entire cohort of 1250 patients was analyzed. Especially given that we tried to evaluate the performance of cytological examination in the entire cohort of urothelial cancer patients (bladder and upper tract) from another perspective (starting from patients with confirmed UTUC during follow-up), see also subchapter “Diagnostic value of cytology of pharmacologically forced diuresis”.

We agree that the number n=1431 might appear confusingly. Given that the majority of patients had several cytological examinations at different time points during the follow-up, a total of 1431 cytology specimens had been analyzed. For ease of understanding, we have adjusted the description in the methods section (see page 3, paragraph 3) and the beginning of the paragraph "Diagnostic value of cytology of pharmacologically forced diuresis", see page 9, paragraph 2.

  1. Furthermore, in cohort one, 25 individuals underwent preoperative cytology examination, and the positive rate obtained, regardless of Stringent or soft criteria, was very low. The reviewer acknowledges the significance of urine cytology analysis of a pharmacologically induced diuresis and is open to accepting both positive and negative results. However, the conclusion cannot be drawn from such a confusing presentation of the data.

Reply: Thank you for this input. We agree with the reviewer regarding the performance of the test. Therefore, we have adapted the conclusion to make this statement more concrete. We have tried to summarize the results schematically and as clearly as possible. Since these are two separate cohorts, of which the second one is analyzed with two different approaches, we substantially reviewed this subchapter in order to make it more understandable, see page 9, subchapter “Diagnostic value of cytology of pharmacologically forced diuresis”.

Reviewer 2 Report

Comments and Suggestions for Authors

The present study confirms and demonstrates the low effectiveness of the forced diuresis method for cytological examination of urine in the diagnosis and follow-up of patients with upper tract urothelial carcinoma. The following aspects require major revision:

 Introduction:

1) Authors use a large number of abbreviations in the text and tables, which makes orientation rather difficult. We suggest to list each abbreviation with its meaning before the body of the text.

2) The data about biomarkers in the introduction might be shortened as it is more appropriate for the discussion section. The authors are encouraged to focus on the study relevance instead.

3) The authors claim the appearance of turbulent urine currents in the upper urinary tract during the forced diuresis, while the study does not describe the mechanism of turbulence. Can the flow of urine be so abundant inside the renal pelvis to cause a swirl of fluid flow in the upper urinary tract?

4) We recommend the authors to transfer information about the patient sample to the “materials and methods” section.

 Materials and Methods:

1)   The authors are encouraged to detain information regarding the study design and selection of patients and the inclusion criteria. If a retrospective study option was chosen, then why were patients given cytology using forced diuresis? Is this a routine practice in the medical institution where the study was conducted? If the study is prospective, how the patients were diagnosed with urothelial carcinoma before enrollment? Did all of them have previous biopsy?

2)   It is mentioned that the patients underwent CT or MRI urography. Could you please specify, how a particular investigation have been chosen? Is there any evidence that their diagnostic accuracy is comparable? This should be supported by a reference.

 Results and conclusion:

1) A comparative analysis using the control group with a conventional cytology  without the forced diuresis method might be reasonable.

2) The authors are encouraged to discuss, whether a collection of 500 ml of urine might introduce a bias? In this way, during centrifugation, it may be possible to obtain a larger number of cells thus increasing the diagnostic accuracy.

Author Response

Reviewer #2

 Introduction: 1. Authors use a large number of abbreviations in the text and tables, which makes orientation rather difficult. We suggest to list each abbreviation with its meaning before the body of the text.

Reply: We are grateful for this objection and completely agree with the reviewer. As recommended, we have removed most of the abbreviations from the text. As for the tables, we have used abbreviations for the purpose of remaining as systematic as possible and have added them as footnotes. However, we can remove the abbreviations from the tables at the discretion of the reviewer if desired.

  1. The data about biomarkers in the introduction might be shortened as it is more appropriate for the discussion section. The authors are encouraged to focus on the study relevance instead.

Reply: We agree with the reviewer and we removed this paragraph from the introduction and have added the content to the discussion section to prevent loss of information, see page 13, paragraph 7.

  1. The authors claim the appearance of turbulent urine currents in the upper urinary tract during the forced diuresis, while the study does not describe the mechanism of turbulence. Can the flow of urine be so abundant inside the renal pelvis to cause a swirl of fluid flow in the upper urinary tract?

Reply: We agree with the reviewer that the mechanical mechanism generated by pharmacologically induced diuresis cannot be valuated with certainty and is furthermore affected by other individual variables for each patient. Nevertheless, it is our collective belief that the rationale behind this mechanism and the biological concept on which it is based is clearly and extensively explained to the reader in the introduction. We have adapted the introduction accordingly, see page 2, paragraph 3. We hope to satisfy the reviewer’s expectation.

  1. We recommend the authors to transfer information about the patient sample to the “materials and methods” section.

Reply: We are grateful for this input. We have deleted most of the patient sample information in the introduction which is now solely explained in the material and method section. We carefully paid attention to prevent repetitive information.

Materials and Methods:

1.The authors are encouraged to detain information regarding the study design and selection of patients and the inclusion criteria. If a retrospective study option was chosen, then why were patients given cytology using forced diuresis? Is this a routine practice in the medical institution where the study was conducted? If the study is prospective, how the patients were diagnosed with urothelial carcinoma before enrollment? Did all of them have previous biopsy?"

Reply: We greatly appreciate this input. We agree with the reviewer that this needs to be specified further. In brief, this is a retrospective study and cytology using forced diuresis has been used as a routine follow-up investigation in our institution. We therefore further specified this information under “Materials and Methods”, see page 3, paragraph 1 and 3.

  1. It is mentioned that the patients underwent CT or MRI urography. Could you please specify, how a particular investigation have been chosen? Is there any evidence that their diagnostic accuracy is comparable? This should be supported by a reference.

Reply: Thank you for this comment. As such, CT-urography was the modality of choice. MR-urography was performed only in case of contraindications for CT, given its slightly lower accuracy. We added an explanation under “Materials and Methods” about the choice of the imaging modality accordingly, see page 3, paragraph

  1. Results and conclusion:

  1. A comparative analysis using the control group with a conventional cytology without the forced diuresis method might be reasonable.

Reply: We thank the reviewer for this comment. In our institution all of the patients were followed-up with pharmacologically forced diuresis. In none of the patients a conventional cytology has been performed. As such, even your suggestion of a comparative analysis would be of great interest, we cannot deliver any data about it. However, we have added it as a limitation to the discussion section, see page 14, paragraph 4.

  1. The authors are encouraged to discuss, whether a collection of 500 ml of urine might introduce a bias? In this way, during centrifugation, it may be possible to obtain a larger number of cells thus increasing the diagnostic accuracy.

Reply: We appreciate this question. In fact, collection and processing of cytology of pharmacologically forced diuresis was not controlled. As such, errors and delay in both steps may be a potential source of bias. Indeed, the number of cells obtained through pharmacologically induced diuresis matters. We can only speculate that pharmacologically forced diuresis obtains a larger number of cells which may increase its diagnostic accuracy. In this context Brimo et al reported that a preliminary cut-off of 10 cells appears to be easily applicable and valid from the clinical standpoint when assigning cases with atypical urothelial cells to the "positive" or the "suspicious" categories. We have added this information to the discussion section, see page 14, paragraph 4.

Round 2

Reviewer 1 Report

Comments and Suggestions for Authors

The paper is well organized and data are presented properly. In my opinion, the paper has been improved and now fulfills the requirements to be published in Cancers.

Reviewer 2 Report

Comments and Suggestions for Authors

The authors addressed all the raised issues. There are no other comments.